# EXCISION SCORE: EVALUATING EDITS WITH SURGICAL PRECISION

## ABSTRACT

Many tasks revolve around editing a document, whether code or text. We formulate the revision similarity problem to unify a wide range of machine learning evaluation problems whose goal is to assess a revision to an existing document. We observe that revisions usually change only a small portion of an existing document, so the existing document and its immediate revisions share a majority of their content.

We formulate five adequacy criteria for revision similarity measures, designed to align them with human judgement. We show that popular pairwise measures, like BLEU, fail to meet these criteria, because their scores are dominated by the shared content. They report high similarity between two revisions when humans would assess them as quite different. This is a fundamental flaw we address.

We propose a novel static measure, Excision Score (ES), which computes longest common subsequence (LCS) to remove content shared by an existing document with the ground truth and predicted revisions, before comparing only the remaining divergent regions. This is analogous to a surgeon creating a sterile field to focus on the work area. We use approximation to speed the standard cubic LCS computation to quadratic. In code-editing evaluation, where static measures are often used as a cheap proxy for passing tests, we demonstrate that ES surpasses existing measures. When aligned with test execution on HumanEvalFix, ES improves over its nearest competitor, SARI, by 12% Pearson correlation and by >21% over standard measures like BLEU. The key criterion is invariance to shared context; when we perturb HumanEvalFix with increased shared context, ES' improvement over SARI increases to 20% and >30% over standard measures. ES also handles other corner cases that other measures do not, such as correctly aligning moved code blocks, and appropriately rewarding matching insertions or deletions.

## 1 INTRODUCTION

Editing is a core skill across countless professions, from writers refining drafts to scientists revising research papers. Example tasks from natural language processing (NLP) include sentiment and style transfer Sudhakar et al. (2019), text simplification Al-Thanyyan & Azmi (2021), grammatical error correction Bryant et al. (2023), and updating factual information Logan IV et al. (2022). Nowhere is this more true than in software development, where code evolves through relentless incremental iteration — bug fixes, optimizations, and feature updates — making precise, efficient editing not just useful but essential. Indeed, many AI4Code tasks boil down to editing code: like automated program repair Monperrus (2023), next edit suggestion Chen et al. (2025), refactoring Pomian et al. (2024), and code commenting Panthaplackel et al. (2020), to name a few.

In this work, we focus on revision tasks, which we define as purposeful edits to a document, whether text or code, that preserve its core semantics. This distinguishes it from rewriting or summarisation, which can fundamentally change a document's thesis or structure. We therefore contend that a revision must, by definition, maintain a high degree of similarity to its source. Operationally, we posit that a revision alters a relatively small portion of a text. While the definition of "small" is necessarily task-dependent, we argue that establishing a practical threshold for tasks is feasible and that larger changes can often be decomposed into a sequence of smaller ones, *aka* revisions.

The LLM tsunami has led to the emergence of edit assistants for both text and code revision. Assessing these assistants introduces the *revision similarity problem*: defining a measure for the similarity of

two revisions of an initial text (or code) that is aligned with human judgement. With such a measure, one can quantify an assistant's performance by how similar its revision is to the reference. For some tasks, building a golden set of references can be prohibitively expensive, calling for a symmetric measure of revision similarity, *i.e.* one that equally weights its input revisions, allowing it to better tolerate a noisy references. Another use for a symmetric measure is clustering revisions. For example, imagine being the maintainer of a Linux kernel subsystem. Rather than manually assess many patches, clustering them by revision similarity and only reviewing representative patches would save time.

Model performance on revision tasks cannot be assessed by humans at scale, so we need an automated measure. Crucially, we need a normalised measure, not a raw distance: if this is not immediate, consider how two operands can be arbitrarily distant in absolute terms, yet arbitrarily similar as a function of their length. Specifically, we want a similarity measure, one that returns a score in $[0..1]$, where $0$ denotes utter dissimilarity and $1$ identity. This measure should be task-agnostic, interpretable, and lightweight.

These three properties rule out dynamic measures, notably pass@$k$, that rely on execution. Their executability constraint is crippling. Even ignoring NLP tasks, many AI4Code tasks do not produce executable code, like code summarization and commit message generation. Even executable code can be nontestable Weyuker (1982). Even considering only code generation, their utility falters in the face of incomplete codebases. Even restricted to tasks that produce testable code, dynamic measures under-approximate program behaviour Dijkstra (1972), which undermines their interpretability, and can be prohibitively computationally expensive. For example, Neubig & Wang (2024) report that evaluation on some 300 samples of SWE-Bench-like dataset took them 2 days; Adamczewski (2025) managed to reduce it to 1 hour per 500 dataset samples with powerful hardware and dedicated containerized environments optimized for the task.[1] Thus, a static measure is an indispensable part of the evaluator's toolbox, necessary for scenarios where dynamic evaluation is infeasible or incomplete.

Existing static measures of textual similarity fall into three categories: lexical, edit-based, and semantic. Lexical measures decompose text into a multiset of predefined features, like $n$-grams, then calculate the similarity of two multisets by atomically comparing their elements. While their $n$-grams do capture local order, they are oblivious to global order. Edit-based measures, in contrast, operate on sequences, so they are inherently sensitive to order. BLEU, ROUGE, Jaccard (adapted to multisets), and TF-IDF are prominent examples of lexical measures. Normalised edit distance built using Levenshtein edit distance is the preeminent edit-based similarity measure. Canonical semantic measures are Word Mover's distance Kusner et al. (2015) and BERTScore Zhang* et al. (2020). These measures struggle with rare words, domain-specific jargon, and nuanced linguistic phenomena like negation and sarcasm. Their scores are often hard to interpret, unlike counting matching n-grams; for example, the difference between scores of 0.7 and 0.8 may not be meaningful or consistent across different models and datasets. When applied to the revision problem, these measures are dominated by the underlying similarity of the revisions and the original text (Section 2).

In this work, we proposing the umbrella term "revision similarity" to unite all ML tasks that can be evaluated by three sequences, an original document and two revisions of it, one a golden reference, and the other, the hypothesis to evaluate. We specify five adequacy criteria that any measure of revision similarity should meet and show how many popular measures fail to meet them. We introduce a new measure — EXCISIONSCORE (ES) — that does. It is a static, task-agnostic, interpretable, and lightweight measure. ES aligns a candidate and a set of reference revisions with their source document to focus on their divergent regions, whose $n$-grams it compares. By constructing its under-approximation from a set of references, ES captures a different subspace of program behavior than dynamic measures, a semantic variability we formalize and explore in Section 3.2.

## 2 STORM CLOUDS IN A BLEU SKY

To assess revision quality, direct comparison seems natural. However, popular pairwise similarity measures, like normalised edit similarity, BLEU, ROUGE, METEOR, and chrF, tend to go wrong

---

[1]Although these estimates include the time needed to run the inference of an LLM, we believe they exemplify the hardships connected with execution-based measures.

because revisions of the same initial version are usually inherently similar, while what matters is comparing their *changes*, not the shared context.

For example, suppose an LLM is asked to replace "**anim**" with "**bar**" and outputs

> *"Lorem ipsum dolor sit amet, consectetur adipiscing elit, sed do eiusmod tempor incididunt ut labore et dolore magna aliqua. Ut enim ad minim ~~veniam~~foo, quis . . . officia deserunt mollit anim id est laborum."*

The LLM clearly failed the task: instead of replacing **anim** with **bar**, it incorrectly substituted **veniam** with **foo**. Most people would consider this edit wrong. Yet, popular pairwise metrics will all score close to their maximum value of 1, contradicting human judgment.

We now generalise this example, then use it to show how BLEU goes wrong in such cases.

**Example 1 (Similar strings).** Let $X$, $Y$, $Z$ and $W$ be strings and let the original sequence be $XY$, the assistant's prediction be $XW$, and the reference revision be $XZ$. Let $ED$ denote edit distance. Now imagine asking a language model to replace $Y$ with $Z$ while keeping the common prefix $X$ but failing and instead replacing $Y$ with $W$. Let us assume

$$ED(Y, Z) = ED(Y, W) = ED(Z, W) = |Y| = |W| = |Z| = l \ll |X|.$$

In this example, the assistant (*i.e.* an LLM) replaced $Y$ with something completely wrong, so we expect a poor score from any measure well-aligned with human intuition. Consider BLEU applied to Example 1, *viz.* $\mathrm{BLEU}(XW, \{XZ\})$. On this example, BLEU's brevity penalty will be 1 and, in the limit as $|X| \to \infty$, the ratios of matched $n$-grams to all $n$-grams in $X$ will go to 1, $\forall n$, so $\mathrm{BLEU}(XW, \{XZ\}) = 1$. In short, although the LLM clearly failed the task, BLEU awards a maximal score to tasks captured by Example 1. All other popular pairwise measures, like normalised edit similarity, ROUGE, METEOR, and chrF, fail in the same way: Like BLEU, in the limit as $|X|$ increases with $l$ fixed, all these metrics go to 1, the perfect match.

We are not the first to observe this problem. When Logan IV et al. (2022) proposed a new benchmark for assessing LLM's ability to make factual updates to text, they observed "**ROUGE is Problematic**. We provide ROUGE F-scores [. . . ] In contrast to the previous results, we find that the simple copy source baseline attains a strong score of 77.4 despite making no updates. [. . . ] This illustrates the importance of evaluating on updates rather than the whole text." The fact that the authors even tried to apply ROUGE to the task where its use is, by their own admission, problematic highlights a blind spot in the community's view of how revision similarity-like problems should be evaluated.

## 3 EXCISIONSCORE: MEASURING REVISION SIMILARITY

Popular pairwise similarity measures fail to solve the revision similarity problem when they are dominated by the shared context inherited from an origin string $O$. We formalize shared context in terms of three-way alignment (Definition 1), then propose *5 Adequacy Criteria*, including invariance to shared context, required for a measure of revision similarity to align with human preference. In Section 3.3, we investigate whether existing measures satisfy these criteria. Finally, we define Excision Score and discuss some of its properties.

### 3.1 CORE CONCEPTS AND UTILITIES

A sequence $s$ is a *revision* of an original document $O$ when $ED(s, O) < \tau$ for some small $\tau$. The specific threshold $\tau$ is task-dependent. Notably, closeness in terms of edit distance implies that a revision is necessarily of similar length. As $\tau$ approaches $\max\{|s|, |O|\}$, the edits become so destructive that the resulting sequence is less of a refinement and more of a new document, even if it retains the original's core semantics, as in the case of summarizing verbose text. We take $A$, $B$ to be revisions of the origin $O$.

Let $\Sigma$ denote the set of tokens our documents consist of. Let $- \notin \Sigma$ be the dedicated gap symbol. Let $\Sigma_- := \Sigma \cup \{-\}$. Let $\Sigma_-^*$ and $\Sigma^*$ stand for the set of finite sequences with and without gaps, respectively.

**Definition 1 (Three-Way Alignment).** For three sequences $A, B, O \in \Sigma^*$, a *three-way alignment* is a rectangular array $R$ of three rows such that (1) each element of $R$ lies in $\Sigma_-$; (2) no column of $R$ consists of gaps only; and (3) $\text{ungap}(R_1) = A$ and $\text{ungap}(R_2) = B$ and $\text{ungap}(R_3) = O$, where $R_i$ refers to the $i$-th row of the array $R$ and $\text{ungap} : \Sigma_-^* \to \Sigma^*$ removes gaps from a sequence.

Unlike the standard definition (Gusfield, 1997, §14), our Definition 1 focuses on a special case of three sequences and explicitly rules out columns with all gaps, which can be useful when studying evolution or as a placeholder for missing data but are meaningless in our setting.

For an alignment table $R$, a column of $R$ is *conserved* if all rows in it are identical. If a column is not conserved, it is *divergent*. A divergent region is, informally, a cluster of adjacent divergent columns.

**Definition 2 (Divergent Region).** For a three-way alignment array $R$, a non-empty sub-array $d$ of $R$ is called a *divergent region* if (1) $d$ has three rows, same as $R$ (it is a mini-alignment table); (2) all columns of $d$ are divergent and contiguous in $R$; and (3) there is no divergent column in $R$ that would be adjacent to $d$ (maximality).

An alignment can yield several, possibly no, divergent regions. Let $\Pi_{\text{div}}(R) = \langle d_1, \ldots, d_k \rangle$ denote the *divergent region projection* that produces the list of the $k$ divergent regions extracted from the alignment of $A, B, O$.

**Example 2.** Let $A = CGTCAA$, $B = CGCACT$, $O = CTGCAATT$. Below is one possible alignment. Although here we are using 4 letters with significance in biology for simplicity, note that alignment can operate at a coarser token level, e.g. $\Sigma = $ English words.

|   | 1 | 2 | 3 | 4 | 5 | 6 | 7 | 8 | 9 |
|---|---|---|---|---|---|---|---|---|---|
| A | C | G | T | – | C | A | A | – | – |
| B | C | – | – | G | C | A | C | T | – |
| O | C | – | T | G | C | A | A | T | T |

In the example, columns 1, 5 and 6 are conserved, the others are divergent. Divergent columns can be understood as atomic edits performed on $O$ by $A$ and $B$. For example, column 2 shows that $A$ and $B$ both decided to remove $G$ from $O$. There are two divergent regions, highlighted by the red rectangles, shown on the right:

$$\Pi_{\text{div}}(R) = \left\langle \begin{vmatrix} G & T & - \\ - & - & G \\ - & T & G \end{vmatrix}, \begin{vmatrix} A & - & - \\ C & T & - \\ A & T & T \end{vmatrix} \right\rangle.$$

## 3.2 Adequacy Criteria for Revision Similarity

Recall that $A$, the reference, and $B$, the hypothesis, are revisions of the original document $O$. We contend that all revision similarity measures should possess the following intuitive properties: (1) Reward edits on $A$ and $B$ agree; (2) Penalize edits on which $B$ disagrees with $A$; (3) Invariant to shared context (matches across all of $A$, $B$ and $O$); (4) Origin-variant ($O$ changing with $A$ and $B$ fixed); and (5) Reward semantically equivalent mismatches.

**Properties 1 and 2:** Rewarding matches while penalizing mismatches is at the core of any ML evaluation task. Revision similarity is no exception, motivating these properties. The word "edits" implies existence of $O$, to which edits are applied, tying them to revision similarity. Despite apparent their simplicity, there are several interesting edge cases, one of which we examplify below.

**Example 3 (Agreeing on Deletions).** Let $D, K, R \in \Sigma^*$ be non-overlapping and assume $O = DKR$ where $D$ is deleted by both revisions, $K$ is kept unchanged and $R$ is replaced. Assume that $A$ and $B$ utterly disagree on what to replace $R$ with, i.e. $A = KR_A$ and $B = KR_B$ with $R_A$ sharing no overlap with $R_B$. Although $A$ and $B$ differ in each replacing $R$ with something different, they do agree on deleting $D$. Therefore a human would expect a partial similarity score.

In Section 2, we illustrated that measures that reward the shared context as match, violating **Property 3**, do not align with human judgement. We rely on the notions three-way alignment (Definition 1) and divergent regions (Definition 2) to formalize invariance to shared context.

**Property 3 (Invariance to Shared Context).** A revision similarity measure $m(A, B; O)$ is invariant to shared context iff $\forall A, A', B, B', O, O' \in \Sigma^*$

$$\Pi_{\text{div}}(A, B, O) = \Pi_{\text{div}}(A', B', O') \implies m(A, B; O) = m(A', B'; O').$$

In words, if the divergent regions of $(A, B, O)$ match those of $(A', B', O')$, a measure $m$ invariant to shared context must return identical scores on those two inputs.

**Property 3** equates shared context with conserved columns. Ignoring shared context, *i.e.* adding or removing conserved columns, is thus equivalent to only considering the divergent regions. A special case of **Property 3** is when $(A, B, O)$ differs from $(A', B', O')$ by a common prefix or suffix. For all sequences $\alpha, \beta$ that do not overlap any of $A, B, O$, measure $m$ must satisfy $m(A, B; O) = m(\alpha A\beta, \alpha B\beta; \alpha O\beta)$.

Another way to conceptualize invariance to shared context would be to constrain the values of $m$ to inputs where the hypothesis revision matches the origin, $B = O$. You can think of this as evaluating a "do-nothing" baseline system that simply echoes the input to produce the output revision. Clearly, such a system should get a bad score, e.g. zero: $m(O, B; O) = 0$. A measure that ignores $O$ has no way of identifying this baseline.

**Property 4 (Origin-variant).** When $A$ and $B \neq A$ be fixed, while we edit $O_1$ to form a sequence of variants $\langle O_1, O_2, \ldots \rangle$, where $\text{ED}(O_i, A) = \text{ED}(O_i, B) < \text{ED}(O_{i+1}, A) = \text{ED}(O_{i+1}, B)$, $m(A, B; O_i) > m(A, B; O_{i+1})$ must hold.

In contrast to **Property 3**, which constrains a measure's handling of added and removed conserved columns in the 3-way alignments, this property concerns changes to $O$'s row. The strict inequality in the variant sequence restricts the changes to conserved columns. Let $l_i = \text{ED}(O_i, B) = \text{ED}(O_i, A)$, as $O$ moves away from $A$ and $B$. We argue that revision similarity should increase along with $l_i$. Indeed, as $O$ becomes more and more distant, $A$ and $B$ are implicitly and independently applying a larger and larger set of matching edits to $O$, increasing their mutual revision similarity.

Finally, **Property 5** introduces dependence on the semantics of the origin document. Most revision similarity tasks admit multiple semantically equivalent solutions. In NLP, syntactic variances are often addressed by providing multiple references that cover different equivalent solutions. However, some semantics-preserving transformations—such as reordering function definitions or inlining—are impractical to enumerate exhaustively in a reference set. A natural alternative is to design the similarity measure itself to tolerate such variations and account for the existence of multiple valid solutions. We now state this property intuitively:

**Property 5 (Obliviousness to Semantically Equivalent Syntactic Variances).** Revision similarity measures should be oblivious to syntactic variances; that is, they should assign the same score when differences arise solely from transformations known to preserve semantics.

For a formal statement and extended discussion, we refer the reader to Appendix A.

### 3.3 THE UNMET NEED FOR ADEQUATE REVISION SIMILARITY METRICS

An intuitive idea for solving the revision similarity problem is to locate and strip out a Longest Common Subsequence (LCS) between the origin $O$ and the two revisions $A, B$ originating from it before applying some pairwise similarity measure. Formally, let us denote the deletion of a subsequence by $\setminus$ and the pairwise similarity measure by $P : \Sigma^* \times \Sigma^* \to [0, 1]$, where $P = 1$ on exactly matching inputs and $P = 0$ if the inputs are utterly dissimilar. Then we define

$$\text{SansLCS}_P(A, B \mid O) \triangleq P(A \setminus L, B \setminus L) \quad \text{where} \quad L = \text{LCS}(A, B, O) \tag{1}$$

Unlike pairwise measures, $\text{SansLCS}_P$ is invariant to shared context being added to $O, A, B$, satisfying **Property 3**. However, $\text{SansLCS}_P$ comes with flaws of its own. By stripping out the LCS *and* considering only $A, B$, we lost the information about what $A$ and $B$ deleted, making it impossible to partially reward agreement on deletions. In Example 3, $\text{LCS}(A, B, O) = K$ and $\text{SansLCS}_P(A, B \mid O) = P(KR_A \setminus K, KR_B \setminus K) = P(R_A, R_B) = 0$. Additionally, $\text{SansLCS}_P$ introduces substring matches that were not possible when comparing $A$ with $B$ directly. By removing the LCS, we introduced n-grams at the junctions that existed in neither $A$ nor $B$ which $P$ might reward, if they happen to match.

A metric named **DiffBLEU** was recently proposed by Bairi et al. (2024); Munson et al. (2022) in the context of code editing and is defined as $\text{BLEU}(\text{diff}(O, B), \text{diff}(O, A))$ where $\text{diff}$ is the output of the $\text{diff}$ program The IEEE and The Open Group (2018) with optional post-processing. Thanks to the clever application of pairwise $\text{diff}$, DiffBLEU adequately addresses the problem of shared content across three revisions. However, lumping deleted and inserted lines together and comparing the concatenated diffs, as opposed to treating them separately, is problematic. First of all, this approach rewards accidental n-gram matches across different operation types. For example, a word inserted by

$A$ would match a word deleted in $O$ by $B$. Such matches do not correspond to what a human would perceive as similarity and thus should not be rewarded. Secondly, while rewarding agreement on deletions by letting BLEU match the deleted lines prefixed by $-$, DiffBLEU is prone to overrewarding it, as the following example shows.

**Example 4 (LHS/RHS Agreement Balance).** Continuing Example 3, suppose that $D$ is empty, so $O = KR$, $A = KR_A$, and $B = KR_B$. Replacements $(R \mapsto R_A)$ and $(R \mapsto R_A)$ can be viewed as two consecutive operations: first deleting $R$, then inserting $R_A$ or $R_B$. In that view, $A$ and $B$ agree on deleting $R$, the left-hand side (LHS) of the replacement, but disagree on $R_A$ versus $R_B$, the right-hand side (RHS). The high BLEU score given to the matching deleted lines will dominate the mismatch between the RHS's of the replacements, contradicting human judgment.

**SARI** (System output Against References and against Input) is a text similarity measure that compares a system's edit (e.g., a simplification) to multiple references and the original input, evaluating the appropriateness of added, deleted, and kept n-grams via precision/recall/F1 scores Xu et al. (2016). SARI is defined as

$$\text{SARI}(I, O, R) = \frac{1}{3}(F_{\text{add}} + F_{\text{keep}} + P_{\text{del}}) \tag{2}$$

where $F_{\text{keep}}$ and $F_{\text{keep}}$ stand for $F_1$ score of the corresponding operation and $P_{\text{del}}$ stands for the precision of deletions, all three averaged over n-grams of order n=1..4. In the limit we described in Example 1, when the shared context dominates, $F_{\text{keep}}$ term of SARI is always greater than $1 - \epsilon$, where $\epsilon \to 0$. This narrows SARI's effective range of values down to $[\frac{1}{3} - \epsilon, 1]$. Although SARI does not fully step into the pitfall of pairwise measures and accounts for $O$, it is not invariant to shared context, failing **Property 3**. In the next subsection we describe EXCISIONSCORE that builds upon SARI and addresses that flaw.

## 3.4 EXCISIONSCORE DEFINED

Armed with the insight that shared context should be removed and sidestepping the mistakes of SansLCS, we define EXCISIONSCORE (ES) as follows:

$$\text{ES}(A, B; O) \triangleq \text{SARI}(A \setminus L, B \setminus L, O \setminus L) \quad \text{where} \quad L = \text{LCS}(A, B, O). \tag{3}$$

After excising the LCS like $\text{SansLCS}_P$ does, ES applies SARI. Recall that $P$ in $\text{SansLCS}_P$ was a pairwise measure, which made it impossible to reward agreement on deletions (Example 3). In contrast, SARI accepts all three documents $A,B,O$ as arguments and has a special term $P_{\text{del}}$ (eq. (2)) dedicated to that. In terms of three-way alignment, removing the LCS can be thought of as extracting divergent regions and concatenating them, then removing the gaps. When converting strings to a set of n-grams, we omit the n-grams that span several divergent regions, sidestepping the other flaw of SansLCS.

EXCISIONSCORE meets the revision similarity adequacy criteria. EXCISIONSCORE identifies edits as kept, added, or removed n-grams, correctly rewarding agreement on deletions, meeting **Properties 1** and **2**. We discussed that due to the $F_{\text{keep}}$ term, SARI is not invariant to shared context and can award a score of up to $\frac{1}{3}$ on the "do-nothing" baseline ($B = O \neq A$). We fix this by excising the shared context and ensuring that the $F_{\text{keep}}$ term does not saturate. Thanks to that, ES correctly returns 0 on the do-nothing baseline and satisfies **Property 3**, if we neglect rare accidental matches that could happen with any n-gram-based measure even when computed on random sequences. ES meets **Property 4**: Changing a shared token in $O$ while keeping $A$ and $B$ fixed turns a previously ignored conserved column into a new divergent region on which $A$ and $B$ agree, increasing $P_{\text{del}}$ and $F_{\text{add}}$ in Equation (2). Finally, ES partially satisifises **Property 5** by matching misplaced insertions, which we found to be a common case in CanItEdit dataset we use in Section 4.

EXCISIONSCORE relies on LCS, which, if computed exactly, implies $\mathcal{O}(l^3)$ time complexity, $l = |O|$. For long $|O|$, this quickly becomes impractical, so we compute $L$ in Equation (3) approximately. In our implementation, $L$ is a not necessarily longest common subsequence computed as $\text{LCS}(\text{LCS}(O, A), \text{LCS}(O, B))$. Two-way LCS computed 3 times brings the time complexity down to $\mathcal{O}(l^2)$.

## 4 EVALUATION: EXCISIONSCORE AS EXECUTION PROXY

Actively developed real-world codebases often include incomplete, non-compilable code for which tests have not yet been written. Even for compilable code in large systems, full build and test cycles can be prohibitively long, making rapid, lightweight static feedback essential. Equipping a dataset with extensive test coverage can be more difficult than mining ground truth solutions. For these reasons, static measures based on syntactic similarity to the reference solution persist as cheap proxies to expensive verification for AI-generated programs. Following widely accepted practice, we therefore explore how well ExcisionScore and other popular static measures correlate with test execution. Our evaluation approach answers the question: "When execution is possible, which static metric best predicts its outcome?".

**Datasets**  We consider two code editing datasets, where each dataset item consists of a code snippet, a natural language edit instruction, a reference solution, and a test suite to verify correctness. **HumanEvalFix** Muennighoff et al. (2023) contains 984 buggy code snippets across 6 programming languages (Python, JavaScript, Java, Go, C++, Rust). **CanItEdit** Cassano et al. (2024) is a dataset of 120 Python programs. The instructions take two forms: "lazy", based on human-authored commit messages, and "descriptive", written by an LLM. In CanItEdit, the LLM is expected to edit, on average, around 21% of the original code in terms of normalized edit distance between the original code snippet and the reference solution. In constrast, expected edits in HumanEvalFix are more constrained (5%) as the bugs are usually small and well-localized. The two datasets also differ in the distribution of ground truth edits. In HumanEvalFix, $|A| \approx |O|$, whereas CanItEdit's references are 20% longer than the original text, indicating prevalence of insertions.

**Experiment Setup**  We obtain 3 LLM outputs for each item of each dataset, using 9 different models to multiply our sample size and the following prompt:

```
Edit the given code according to the instructions.
You MUST output the complete revised version of the code with your edits.
You MUST NOT add any comments. DO NOT explain you edits.
## Instructions
{instruction}
## Input code
{input_code}
## Edited code:
```

The LLMs used are: `claude-sonnet-4` Anthropic (2025), Google's `gemini-2.5-flash` DeepMind (2025), OpenAI's `gpt-4o-2024-11-20`, `gpt-4o-mini-2024-07-18`, `gpt-4.1-nano-2025-04-14` OpenAI (2025), Qwen2.5-Coder Instruct 1B and 5B Hui et al. (2024), DeepSeek Coder Instruct 1.3B and 6.7B Guo et al. (2024). For the Qwen and DeepSeek models, we use vLLM inference engine Kwon et al. (2023) and the default sampling parameters. For the remaining (proprietary) models, we set temperature to 0.2 and `top_p` to 0.95.

This results in 26568 ($3 \times 9 \times 984$) data samples derived from **HumanEvalFix** dataset and 2835 ($3 \times 9 \times 105$) derived from **CanItEdit**. For each LLM output, we execute the tests and record a binary pass (1) or fail (0) score. In HumanEvalFix, 45% of the generated solutions pass the test, while for CanItEdit dataset this number is 40%. Finally, we report Pearson correlation coefficient between the 0/1 indicator of passing the test and ES along with various other static measures computed on the (origin, reference, prediction) triples, namely exact match, unnormalized Levenshtein distance (ED), NES, chrF, BLEU, CodeBLEU, DiffBLEU, and SARI.

We experiment with 2 implementations of ExcisionScore—ES-Line and ES-Token—differing in the granularity of LCS. ES-Line excises the shared lines of code, while ES-Token tokenizes the code strings with `tree-sitter` and excises tokens common to all three strings. Additionally, we remove comments that do not affect execution, before passing $A, B, O$ to each measure.

To illustrate what happens if our datasets contained a larger proportion of shared context, we artificially expand it by prepeding a long shared prefix of random length to each $A, B, O$, similar to Example 1. Since the measures considered are semantics-agnostic, the exact content of the prefix is irrelevant. The prefix is sampled uniformly from characters `abcdef`, a whitespace, and a newline character to contain a total of 2000–3000 characters. A different prefix is generated for each individual dataset sample. We re-use the unperturbed pass/fail test execution data and compute the Pearson correlation coefficients. After these perturbations, the reference solution's coverage drops, on average, from 21% to 7% of the original code in CanItEdit and from 5% to only 0.5% in HumanEvalFix.

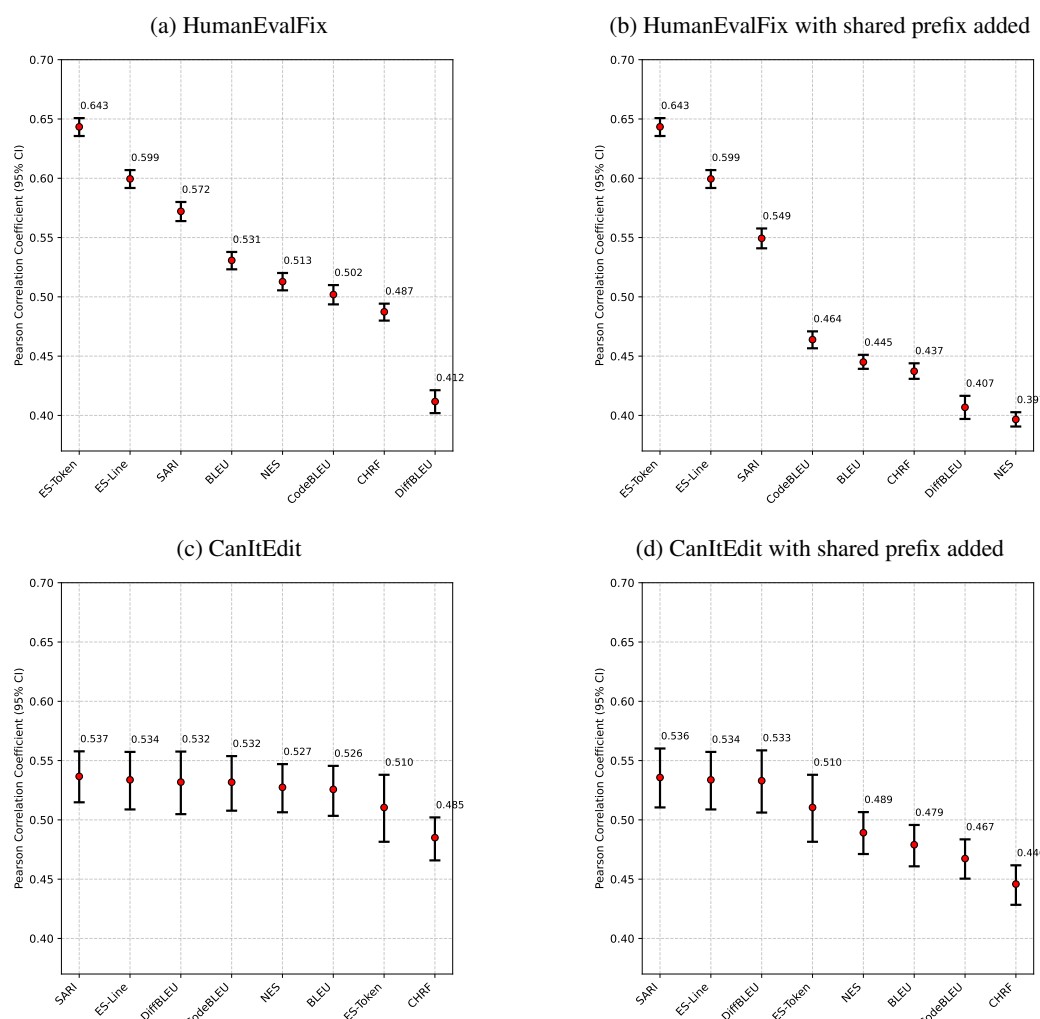

Figure 1: Correlation of various static measures with test execution. The first row refers to HumanEvalFix dataset, and the second to CanItEdit. Error bars indicate 95% bootstrap confidence intervals. The plots in the right column pertain to the experiment where we prepend a large random prefix to $A, B, O$. We excluded ED and exact match as their coefficients were low, as expected.

**Results** The resulting correlation coefficients are in Figure 1. On **HumanEvalFix** dataset, ES-Token takes the lead with a correlation coefficient of $r = 0.643$ (CI $[0.636, 0.651]$), followed by ES-Line $r = 0.599$ (CI $[0.592, 0.607]$), SARI $r = 0.572$ (CI $[0.564, 0.58]$), and others. Both ES-Token and ES-Line offer statistically significant improvement upon SARI, indicating that it is beneficial to remove shared context before applying SARI. When shared context dominates, the $F_{\text{keep}}$ term of SARI is always maxed out to 1. Intuitively, that means that one of the 3 degrees of freedom ($F_{\text{keep}}, F_{\text{add}}, P_{\text{del}}$) SARI has is permanently switched off, making SARI less sensitive. When a shared context is added to the HumanEvalFix dataset, SARI's correlation coefficient with pass@1 drops significantly: from 0.572 (CI $[0.564, 0.58]$) to 0.549 (CI $[0.541, 0.558]$). Extending shared context does not affect SARI's correlation with test execution on CanItEdit.

On the **CanItEdit** dataset, the differences in performance of different metrics, including pairwise ones, are insignificant. One possible explanation for that is the relative size of the edited region in CanItEdit (21%, not including the unexpected edits the LLM solution makes). Besides, CanItEdit expects 20% more insertions than deletions. Since the inserted tokens appear in either $A$ or $B$ but not in $O$, the benefits of taking $O$ into account are reduced.

Our results indicate that granularity of computing LCS or alignment is important. In HumanEvalFix, the edits are often small, changing only a few tokens within a line, explaining why ES-Token surpasses ES-Line on this dataset. On CanItEdit, however, ES-Token loses to ES-Line by a barely significant margin. Manual inspection reveals that overly fine-grained alignment can lead to meaningless unintuitive artifacts. Similarly, line-granular DiffBLEU falls short on HumanEvalFix, while performing well on CanItEdit.

We found that 2% of LLM solutions for HumanEvalFix dataset and 5% for CanItEdit match the reference solution exactly. Excluding them from the data decreases all the correlation coefficients by 3-9%, leaving the ranking of different metrics unaffected on HumanEvalFix. Differences in correlation coefficients remain insignificant on CanItEdit.

Perturbing the data by adding a shared prefix does not affect ES and DiffBLEU scores, as ignoring this prefix was part of their design, neither does it affect unnormalized ED. In contrast, correlation coefficients of other pairwise measures with pass@1 on HumanEvalFix drop significantly: from $[0.505, 0.52]$ to $[0.439, 0.451]$ for BLEU, from $[0.494, 0.51]$ to $[0.457, 0.471]$ for CodeBLEU, from $[0.48, 0.494]$ to $[0.431, 0.444]$ for chrF, and from $[0.505, 0.52]$ to $[0.391, 0.403]$ for NES. We observe a similar effect on CanItEdit.

We critisized pairwise measures on the grounds that they reward dominating shared context as match, which reduces the effective range of values the similarity measure can take from $[0, 1]$ to $[x, 1]$, where $x$ depends on how prevalent shared context is in the data. Some might object to this argument and suggest that dataset-specific re-normalization of the scores could be a trivial remedy. Namely, if $s_i$ are the pairwise similarity scores on the $i$-th dataset sample, one could consider $s_i' \equiv \frac{s_i - \min s_i}{\max s_i - \min s_i}$, ensuring that $s_i'$ fully cover the expected $[0..1]$ range. However, our empirical results suggest that such a re-normalization still does not yield a satisfactory measure. Pearson's correlation coefficient $r$ is invariant under linear transformations of the variables. Thus, the renormalized scores would have the same low $r$ as the original values we observed. Our superiority argument based on correlation coefficients holds independently of the arguments about suitable and interpretable range of values.

## 5 RELATED WORK

Our adequacy criteria leverage global multiple sequence alignment (MSA, Definition 1), a technique well established in bioinformatics Chatzou et al. (2015).

Numerous measures assess the similarity of two strings in different contexts. We argue that they are ill-suited for the revision similarity problem, because they do not take the original document $O$ string into account. Without $O$, pairwise measures cannot distinguish between revision similarity due to inheriting parts of $O$ unchanged and that due to performing the same edits to $O$. As a result, pairwise measures reward shared context. Pairwise N-gram-based lexical measures include BLEU Papineni et al. (2002), METEOR Banerjee & Lavie (2005), ROUGE Lin (2004), and chrF Popović (2015). BLEU has well-documented limitations, including its inability to address revision similarity — a gap we rigorously analyze in Section 2. For a systematic critique of BLEU's shortcomings (*e.g.*, its insensitivity to paraphrasing and granular edits), we direct readers to Callison-Burch et al. (2006) and Reiter (2018). Despite these flaws, BLEU persists as a de facto standard due to its simplicity, reproducibility, and historical inertia. Syntax-aware pairwise static measures for code include Tree Edit Distance Schwarz et al. (2017) (TED), RUBY Tran et al. (2019), CodeBLEU Ren et al. (2020), and CrystalBLEU Eghbali & Pradel (2023). They require the code to be parseable. TED is very interpretable but takes $\mathcal{O}(n^3)$ time, where $n$ is the number of nodes in the syntax tree. CrystalBLEU removes 'shared' n-grams, but interprets 'shared' as frequent n-grams in the language, not unedited tokens inherited from $O$ as in our work. Lastly, embedding-based measures such as cosine similarity, BERTScore Zhang et al. (2020) and CodeBERTScore Zhou et al. (2023), while capturing semantic similarity, remain fundamentally pairwise and cannot account for the original document $O$.

Evaluating Grammatical Error Correction (GEC) techniques can be seen as an instance of the revision similarity problem. In GEC, edits are typically small and evaluation requires strict measures that are sensitive to word order. Alignment has been used to leverage these aspects in designing GEC metrics such as I-Measure Felice & Briscoe (2015) and $M^2$ (MaxMatch) Dahlmeier & Ng (2012). Evaluation of Text Simplification (TS) can, in some cases, also be viewed as revision similarity

problem, provided that the simplifying changes do not rewrite the entire text. SARI Xu et al. (2016) (defined in Equation (2)) is an n-gram-based metric designed for text simplification.

## 6 REPRODUCIBILITY STATEMENT

We attach the collected LLM responses and test execution results as Supplementary Materials. In Section 4 we specify how it was obtained. The code used to process this data and compute the scores with their correlation coefficients is available anonymously under https://anonymous.4open.science/r/excision-score-eval-B9AF/ .

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

## A  FORMALISM AND EXTENDED DISCUSSION OF PROPERTY 5

Standard reference-based evaluation relies on a set of references $\mathcal{A}$ to define the space of semantically acceptable revisions for a given task, a well-established practice in fields like Grammatical Error Correction (GEC) to account for valid alternative outputs (Bryant et al., 2017, §6). For instance, to evaluate a revision that adds a sort function, $\mathcal{A}$ would contain different correct sorting algorithms (e.g., quicksort, mergesort), thereby defining a range of semantic solutions deemed acceptable.

However, a single semantic solution can often be implemented with minor syntactic variances that do not alter its meaning. Examples include moving a function definition to a different, syntactically valid location within a code file, or applying semantics-preserving code transformations such as inlining (replacing a function call with its body) or outlining (extracting a code segment into a new function). In text, analogous examples include adding a new step before any of its dependencies or changing the order of a bulleted list.

Requiring $\mathcal{A}$ to explicitly enumerate all possible variants of every acceptable solution is computationally expensive. For instance, to handle all possible locations for a single function definition across $\mathcal{O}(|O|)$ locations, one would need a reference for each. Comparing a candidate revision against this set involves a quadratic $\mathcal{O}(|O|^2)$ number of pairwise block comparisons to align the candidate's function with each reference's function. This cost is incurred for each such syntactic variance.

Therefore, a robust similarity measure should be efficient with respect to the reference set $\mathcal{A}$ for such semantics-preserving, syntactic variances. **Property 5** requires that the measure, leveraging knowledge of the language and task, can recognize a candidate revision $B$ as matching a reference $A \in \mathcal{A}$ even if $B$ and $A$ differ only by a variance of this kind, without requiring $\mathcal{A}$ to explicitly enumerate all possible variants. Moreover, the measure should acknowledge a *partial* match in case some of the edits $B$ makes achieve the effect semantically equivalent to some of $A$'s edits, even if $A$ otherwise semantically differs from $B$.

**Example 5 (Tolerated Syntactic Variances).** In the sort function task, the set $\mathcal{A}$ defines the acceptable semantic range (e.g., quicksort, mergesort). **Property 5** ensures that if a candidate revision $B$ places a correct quicksort implementation in a different location than the reference quicksort in $\mathcal{A}$, it is still correctly identified as matching the quicksort semantics. This avoids the need for a separate reference for every possible function location, making the evaluation both practical and semantically grounded.

To formalize the notion of semantics-preserving syntactic variances, we first define a semantic equivalence relation between documents under a set of semantics-preserving transformations.

**Definition 3 (Equivalence under Semantics-Preserving Transformations).** Let $[\![s]\!]_L$ denote the semantics of a string $s$ in language $L$. Let $\equiv_L$ be a semantics-preserving syntactic equivalence relation on strings in $L$, such that $A \equiv_L B$ implies $[\![A]\!]_L = [\![B]\!]_L$. This relation is defined by a set of syntactic transformation rules (e.g., function relocation, inlining) that are known *a priori* to preserve semantics. Any practical measure uses a concrete $\equiv_L$ that under-approximates the full, undecidable semantic equivalence relation.

We now extend this concept to define semantic equivalence between *sets of edits*.

An *atomic edit* is a tuple defining a single, irreducible change to a sequence, *e.g.*, of tokens or characters. It combines an operation (insert, delete, replace, swap, *etc.*), a target location index within the sequence, and an optional operand: a new element for insert/replace, an index for swap, ignored for delete. Let $A$ be the revision produced by applying a set of atomic, nonoverlapping edits $\{a_1, \ldots, a_n\}$ to $O$, denoted $A = \{a_1, \ldots, a_n\}(O)$. Similarly, let $B = \{b_1, \ldots, b_m\}(O)$. By nonoverlapping, we mean that the indices of distinct edits are distinct: formally, for $a_i = (\_, x, \_), a_j = (\_, y, \_) \in \{a_1, \ldots, a_n\}, i \neq j \Rightarrow x \neq y$, where '$\_$' denotes don't care. The edits in a nonoverlapping set can be applied simultaneously. For any subset of indices $I \subseteq \{1, \ldots, n\}$, we denote by $a_I(O) = \{a_i\}_{i \in I}(O)$ the revision obtained by applying exactly those edits to $O$. For example, $\{\}(O) = O$ and $\{a_i\}_{i=1}^{n}(O) = A$.

**Definition 4 (Semantically Equivalent Edits).** For $\{a_i\}_1^n$ and $\{b_i\}_1^m$, let $I \subseteq 1..n$, $J \subseteq 1..m$, $\bar{I} = 1..n \setminus I$, $\bar{J} = 1..m \setminus J$. Assume that the indices of the edits $b_J$ do not overlap with the indices of $a_{\bar{I}}$ and indices of $a_I$ do not overlap with those of $b_{\bar{J}}$.

The subsets of edits $a_I = \{a_i \mid i \in I\}$ and $b_J = \{b_j \mid j \in J\}$ are **semantically equivalent under** $\equiv_L$ iff:

$$A \equiv_L (b_J \cup a_{\bar{I}})(O) \quad \wedge \quad B \equiv_L (a_I \cup b_{\bar{J}})(O)$$

This relation is symmetric. It formalizes the condition that the edits in $a_I$ and $b_J$ are interchangeable syntactic variances for achieving the same semantic outcome within the context of their respective revisions; in other words, the edits are equivalent in the surrounding context of the other edits $A$ and $B$ apply. Finding the subsets $I$ and $J$ is, of course, undecidable in general, but, in practice, can be under-approximated with knowledge of the transformations that underlie a particular concretisation of $\equiv_L$.

**Property 5 (Obliviousness to Semantically Equivalent Syntactic Variances).** In the notation of Definition 4, a similarity measure $m$ must satisfy:

$$m(A, B; O) = m(A, (a_I \cup b_{\bar{J}})(O); O),$$

whenever the edits $\{a_i \mid i \in I\}$ and $\{b_j \mid j \in J\}$ are semantically equivalent under $\equiv_L$.

This property requires a revision similarity measure to be oblivious to the choice between syntactically different but semantically equivalent edit sets under some realisation of $\equiv_L$. Our new measure, EXCISIONSCORE does so by being insensitive to the order of n-grams, as we show in Section 3.4. This accounts for semantically equivalent mismatches that can be fixed by reordering blocks of content, which we found to be common in our our evaluation datasets.

Humans care about more than mere semantics when comparing revisions. For instance, a human may prefer unobfuscated or well-refactored code or, in text, a paraphrase or a summary of some topic. Humans, to take another example, also vary greatly in terms of their code commenting preferences. **Property 5** permits handling these aspects in two ways: either by defining $\equiv_L$ to consider them or by including examples of these aspects in the set of references. Using $\equiv_L$ to do so effectively makes the relevant aspects semantic.

