# OpenReview forum: "Excision Score: Evaluating Edits with Surgical Precision"
_ICLR.cc/2026/Conference — ICLR 2026 Conference Desk Rejected Submission_

### Official Review · Reviewer_9vsX · 2025-10-29

**Soundness:** 3
**Presentation:** 3
**Contribution:** 3
**Rating:** 6
**Confidence:** 4

**Summary:**

This paper presents a metric for evaluating revisions to existing documents. The basic idea is to use LCS to calculate both the deviation of the generated revision to the target document and the deviation of the ground truth revision to the target document, and compare how the two deviations differ from each other. The empirical evidence indicates that the metric proposed in this paper has better correlation with the results of dynamic execution for two code editing tasks.

**Strengths:**

1. The idea of using LCS to establish a metric for revision tasks is novel.
2. The empirical results can confirm the effectiveness of the proposed metric.

**Weaknesses:**

1. Since the correlation is based on comparing each metric with actual execution, it seems that actual execution is a perfect metric. Thus, the usefulness of the proposed metric is mainly due to its efficiency or circumstances where execution is infeasible.
2. The impact of exact matches is not considered in the evaluation. Since the metric of Exact Match can already identify generated revisions that are identical to ground truth revisions, it should be more interesting to show the effectiveness of each metric on only cases where generated revisions literally differ from ground truth revisions.
3. NLP revision tasks may significantly differ from code revision tasks. Therefore, I feel that it may be an over claim that the proposed metric is also effective for NLP revision tasks. Or some empirical evidence should be provided.

**Questions:**

1. If we do not consider the difference in efficiency, what are thhe circumstances where ES is clearly superior to actual execution?
2. If we remove the cases where the generated revisions are identical to the ground truth revisions, how would the correlation results become?
3. Have you experimented on revision tasks in the domain of NLP?

---

> ### Author Response · Authors · 2025-11-26
>
> We are grateful for the structured review and thoughtful questions; we have actioned the exact match concern and have devised an experiment to address the code bias you pointed out.
>
> **Weakness 1 & Question 1: On the relationship between ES and actual execution.**
>
> The reviewer raises a crucial point. We agree that test execution is an effective proxy for functional correctness. However, it is fundamentally an under-approximation --- a program that passes available tests may still contain bugs, and test suite quality varies dramatically.
>
> Our goal is not to claim ES is superior to execution where execution is feasible and perfect, but to position ES as a complementary, static proxy that is valuable when execution is:
>
> - Infeasible or Impractical: As noted in our response to Reviewer C6eL, this includes large codebases with long build/test cycles (e.g., OS kernels) and, most importantly, code under active development that is syntactically incomplete or invalid. You cannot execute a half-written function.
>
> - Inadequate: As noted in our response to Reviewer dBTr30, even when tests exist, they are rarely exhaustive. A metric like ES, which measures the structural similarity of an edit to a ground-truth fix, can capture human expectations like maintainability, style, and intent. It provides a different, valuable signal.
>
> In short, ES is superior to execution in the specific circumstances where execution either cannot be performed at all --- that is, on incomplete or non-compilable code during the editing process itself --- or does not fully capture human preference. Both are common scenarios in real-world development that dynamic evaluation cannot address.
>
> **Weakness 2 & Question 2: On the impact of Exact Matches.**
>
> This is an excellent suggestion for a more granular analysis. We found that 2% of solutions in our HumanEvalFix dataset and 5% in the CanItEdit dataset are matching the reference exactly. Excluding exact matches (EM) from the dataset, we computed the same Pearson correlation of ES and other measures with test execution.
>
> Removing exact matches lowers correlation scores on HumanEvalFix, as metrics no longer benefit from cases where they aligned perfectly with execution results. However, the overall ranking stays the same: ES-Token still clearly leads SARI. On CanItEdit, correlations also drop, and the ranking shifts slightly, but differences among the top five metrics remain negligible. We conclude that presense of exact matches does not affect the qualitative result of the experiment. We will include this observation in the revised manuscript.
>
> **Weakness 3 & Question 3: On applicability to NLP tasks.**
>
> The reviewer is correct that our empirical validation is a coding task. However, the core problem ES solves --- metrics being dominated by shared context --- is universal to revision tasks. For example, in text simplification, a model might change only a few complex phrases in a long paragraph.
>
> We can add a small-scale qualitative analysis of its behavior on a text revision dataset (e.g., XATU [1]) in the appendix to illustrate its general principles, if the reviewers find it beneficial. We would be grateful if the reviewers could direct us to any existing human-labeled text revision datasets that may be relevant or suggest another way to bypass the burden of manually annotating such a dataset.
>
> [1] Zhang, Haopeng, et al. "XATU: A Fine-grained Instruction-based Benchmark for Explainable Text Updates." Proceedings of the 2024 Joint International Conference on Computational Linguistics, Language Resources and Evaluation (LREC-COLING 2024). 2024.

---

> > ### Author Response · Authors · 2025-11-26
> >
> > Below we attach the numeric results for the Exact Match experiment. On HumanEvalFix dataset:
> >
> >     ┏━━━━━━━━━━━━━━━┳━━━━━━━━━━━━━━━━━━━━━━━━━━┳━━━━━━━━━━━━━━━━━━━━━━━━━┓
> >     ┃ Metric        ┃ Pearson 95% CI (with EM) ┃ Pearson CI (without EM) ┃
> >     ┡━━━━━━━━━━━━━━━╇━━━━━━━━━━━━━━━━━━━━━━━━━━╇━━━━━━━━━━━━━━━━━━━━━━━━━┩
> >     │ ES-Token      │ 0.643   [0.636, 0.651]   │ 0.634  [0.626, 0.641]   │
> >     │ ES-Line       │ 0.599   [0.592, 0.607]   │ 0.588  [0.58, 0.595]    │
> >     │ SARI          │ 0.572   [0.564, 0.58]    │ 0.56   [0.551, 0.567]   │
> >     │ BLEU          │ 0.531   [0.523, 0.538]   │ 0.522  [0.515, 0.529]   │
> >     │ NES           │ 0.513   [0.505, 0.52]    │ 0.503  [0.496, 0.511]   │
> >     │ CodeBLEU      │ 0.502   [0.494, 0.51]    │ 0.49   [0.481, 0.498]   │
> >     │ CHRF          │ 0.487   [0.48, 0.494]    │ 0.479  [0.472, 0.486]   │
> >     │ DiffBLEU      │ 0.412   [0.402, 0.421]   │ 0.39   [0.38, 0.4]      │
> >     └───────────────┴──────────────────────────┴─────────────────────────┘
> >
> > On CanItEdit dataset:
> >
> >     ┏━━━━━━━━━━━━━━━┳━━━━━━━━━━━━━━━━━━━━━━━━━━┳━━━━━━━━━━━━━━━━━━━━━━━━━┓
> >     ┃ Metric        ┃ Pearson 95% CI (with EM) ┃ Pearson CI (without EM) ┃
> >     ┡━━━━━━━━━━━━━━━╇━━━━━━━━━━━━━━━━━━━━━━━━━━╇━━━━━━━━━━━━━━━━━━━━━━━━━┩
> >     │ NES           │ 0.527    [0.506, 0.547]  │ 0.491   [0.47, 0.513]   │
> >     │ BLEU          │ 0.526    [0.503, 0.546]  │ 0.49    [0.468, 0.511]  │
> >     │ CodeBLEU      │ 0.532    [0.508, 0.554]  │ 0.487   [0.462, 0.51]   │
> >     │ SARI          │ 0.537    [0.515, 0.558]  │ 0.487   [0.46, 0.511]   │
> >     │ ES-Line       │ 0.534    [0.509, 0.557]  │ 0.484   [0.455, 0.511]  │
> >     │ DiffBLEU      │ 0.532    [0.505, 0.558]  │ 0.474   [0.445, 0.502]  │
> >     │ CHRF          │ 0.485    [0.466, 0.502]  │ 0.454   [0.434, 0.473]  │
> >     │ ES-Token      │ 0.51     [0.482, 0.538]  │ 0.449   [0.42, 0.481]   │
> >     └───────────────┴──────────────────────────┴─────────────────────────┘

---

### Official Review · Reviewer_C6eL · 2025-10-30

**Soundness:** 3
**Presentation:** 4
**Contribution:** 3
**Rating:** 6
**Confidence:** 4

**Summary:**

This paper introduces Excision Score, a new static metric designed for the revision similarity problem. This problem involves evaluating generated document edits against a ground-truth revision, given the original source. The authors argue that popular metrics like BLEU are ill-suited for this, as their scores get inflated by the large amount of unchanged text shared between the original and revised versions. ES solves this by first removing the shared content (identified via a 3-way Longest Common Subsequence) from the original, the reference, and the prediction. It then applies the SARI metric only to the remaining parts where the text actually differs, allowing it to focus on the quality of the edit itself. In experiments on code-editing datasets, ES shows a significantly higher correlation with test execution pass/fail rates than existing metrics like SARI and BLEU, especially on edit-heavy scenarios. The paper further demonstrates ES's robustness by showing its performance is unaffected by adding large amounts of shared, irrelevant context, a scenario where other metrics degrade.

**Strengths:**

- A well-motivated problem: The paper clearly defines the task of revision similarity and highlights a fundamental flaw in using standard metrics like BLEU for it. As AI-driven editing becomes more common, a reliable metric for this is highly important.

- An intuitive and well-motivated solution: The core idea of removing the shared context to focus on the edited regions is simple, elegant, and directly addresses the stated problem. The authors also justify their design by showing how ES avoids the specific pitfalls of related metrics like SARI and DiffBLEU.

- Strong results: The evaluation shows ES has clear advantages. On the HumanEvalFix dataset, ES-Token's correlation with test pass/fail rates was significantly better all baseline methods. A stress test that adds a large shared prefix to all inputs confirmed the metric's robustness over other baseline methods.

**Weaknesses:**

- Limited Semantic Understanding: The metric is still fundamentally lexical. Since the final step uses SARI, the evaluation relies on matching n-grams within the edited regions. The paper itself acknowledges this limitation, stating that ES only "partially satisfies" its own criterion for semantic equivalence (Property 5) . While this approach is shown to handle simple cases like misplaced insertions, it would likely fail to reward more complex, semantically-equivalent-but-lexically-different code (e.g., refactoring a for loop to a list comprehension) if the n-grams don't align.

**Questions:**

The "LLM-as-a-judge" paradigm is a popular evaluation approach nowadays. Have you considered comparing your approach with an LLM-based baseline (e.g., using a small Qwen model to score the edit's correctness) to see which metric achieves higher agreement with human judgment or results? I am curious if there's a model performance threshold above which an LLM assessment becomes more accurate or correlative than static, n-gram-based metrics like ES.

---

> ### Author Response · Authors · 2025-11-26
>
> We thank the reviewer for their feedback; you have helped us improve our discussion of the property 5 and we plan to implement your LLM-as-judge suggestion.
>
> **W1: Limited Semantic Understanding:**
>
> You are right to point out that ES is fundamentally syntactic and has limited semantic understanding.
>
> Nevertheless, like many other syntactic metrics, ES still can approximate semantics by relying on the judicious selection of reference solutions. For instance, in a task like implementing a sort function, one could use a set of reference implementations (e.g., quicksort, mergesort, etc.). One could then compute ES against each and report the maximum score, accounting for the plurality of possible semantically equivalent solutions.
>
> More crucially, our primary target for coding tasks is evaluating revisions to under-tested, non-executable, or practically un-testable code. The latter includes massive codebases where a full build or test suite run is prohibitively expensive. For example, making a localized edit to the Chromium browser, the Linux kernel, or a large CAD/CAM application could require hours to rebuild and test. During active development---especially when a developer is iterating on a new feature---most intermediate revisions are syntactically invalid, and even valid ones cannot be dynamically tested after every minor change. It is precisely in these contexts, where rapid, lightweight feedback is essential, that dynamic evaluation fails and flawed static metrics like BLEU have persisted. For this widespread and practical problem, ES provides a significantly more reliable static proxy.
>
> In light of your feedback, we will refine the formulation of P5 to make explicit how it can leverage multiple references in the revised manuscript. Your comment has also prompted us to consider to enriching ES, similar to CodeBLEU [1], with structural understanding by injecting AST match and some semantics via data-flow. Analogous to how BLEU serves as the basis for CodeBLEU, we view ES as an initial, syntactic foundation for future metrics that incorporate semantic information. It is interesting to note, however, that, despite underapproximating semantics with a set of references, ES outperforms CodeBLEU in our experiments.
>
> **Q1: LLM-as-a-Judge**
>
> This is an excellent suggestion. We agree that comparing ES to an LLM-as-a-Judge baseline would be a valuable experiment clearly sitating ES's performance w.r.t. this increasingly popular evaluation paradigm.
>
> We could
>
> 1. Take a smaller representative sample from our HumanEvalFix dataset;
> 2. Ask the the LLM judge to predict the test execution result for the LLM-generated solutions from that dataset;
> 3. Compute its Pearson correlation coefficient with the actual test execution results;
> 4. Compare the LLM-as-a-Judge's correlation with the correlation coefficients for ES and other static measures;
> 5. To address the matter of "model performance threshold above which an LLM assessment becomes more accurate", repeat 1-4 with two LLMs: a "small" one (e.g. Qwen3-Coder-30B-A3B-Instruct) and a "large" one (e.g. GPT-5).
>
> Prior to conducting the experiment, we would expect ES's correlation coefficient to fall between the one produced by the small LLM judge and the one produced by the "large" one. It has been reported [2] that "small LLMs struggle in judging tasks" and "even GPT-4-turbo frequently fails in assessing code correctness".
>
> Does the proposed experiment design align well with your initial vision? Do you think including it would strengthen the paper?
>
> [1] Ren, Shuo, Daya Guo, Shuai Lu, Long Zhou, Shujie Liu, Duyu Tang, Neel Sundaresan, Ming Zhou, Ambrosio Blanco, and Shuai Ma. "CodeBLEU: a method for automatic evaluation of code synthesis." arXiv preprint arXiv:2009.10297 (2020).
>
> [2] Z. Gao et al., On the Effectiveness of LLM-as-a-Judge for Code Generation and Summarization. arXiv:2507.16587, 2025. Available at: https://arxiv.org/pdf/2507.16587 .

---

### Official Review · Reviewer_dBTr · 2025-10-30

**Soundness:** 1
**Presentation:** 1
**Contribution:** 1
**Rating:** 0
**Confidence:** 4

**Summary:**

This paper proposes a static similarity measure (ES) to compare a proposed solution (S) that is the revision of original document (O) with a reference solution (S'). This specific measure aims to satisfy four properties: reward matches, penalize mismatches, being invariant to shared context (anything unchanged should be ignored), and being invariant to O changing, and aims to reward semantically equivalent mismatches.

The measure consists of applying another measure (SARI) after removing the longest common subsequence.

The evaluation compares other measures (like SARI) to ES by analyzing their correlation to execution match.

**Strengths:**

- Evaluating the correctness of code is an important problem, and it is indeed not always possible to execute code.
- The paper analyses highlights limitations of existing token-based metrics, and focusing on the *changed* parts of code is a sensible observation.

**Weaknesses:**

- The paper makes very strong claims about execution not being a suitable measure for revision tasks, yet considers approximation of execution as the only metric in its evaluation. Extensive benchmarks like SWE-Bench can take a long time to evaluate, but a significant portion of this time is the time to obtain a solution, not just the time to evaluate the solution. I would argue that under-approximating program behavior using execution is better than not even evaluating proper syntactic correctness: it seems better to evaluate whether a program yields *a* correct solution than not yield a compilable or executable solution at all. The two benchmarks used for evaluation actually do facilitate easy execution, further decreasing the motivation for this paper.
- The only property that the proposed solution solves with respect to previous solutions is invariance to shared context. Besides being an incremental improvement, it solves the wrong problem for code: the semantic correctness property.
- A common approach to measure similarity between code is using tree edit distances [1] on abstract syntax trees. Whereas not addressing semantic correctness, it does require code to be syntactically correct. Tree edit distances are completely ignored.
- Semantic scores (like CodeBERTScore [2]) are completely ignored in the evaluation.

[1] http://tree-edit-distance.dbresearch.uni-salzburg.at/
[2] Zhou, S., Alon, U., Agarwal, S., & Neubig, G. CodeBERTScore: Evaluating Code Generation with Pretrained Models of Code. In The 2023 Conference on Empirical Methods in Natural Language Processing.

**Questions:**

Did you encounter any real-world uses for this evaluation metric that is not solved by existing measures, like tree edit distances or semantic similarities?

---

> ### Author Response · Authors · 2025-11-26
>
> We thank the reviewer for their feedback and address the points raised.
>
> **On ES versus Execution.**
>
> The reviewer claims that execution is a better approximation than a static metric, even for uncompilable code. This is a logical impossibility: **you cannot execute uncompilable code.** Static metrics are most valuable precisely when dynamic metrics fail completely --- when no program can be executed. This case is common: developers often work with incomplete, uncompilable code, frequently without tests. Some code is inherently untestable (Weyuker, 1982). Even testable code may have poor coverage, and test oracles can be wrong. Execution is not always available or reliable. Static metrics exist because they remain useful in these settings; BLEU persists for this reason.
>
> Even when execution is possible, functional correctness is not a full proxy for human preference. In AI-based code editing, humans care about clarity, maintainability, idioms, and intent [7]. LLMs frequently violate these expectations: unnecessary comments, outdated conventions, unreadable one-liners, or trivial syntax errors that make otherwise good edits unparsable. Conversely, a model may generate code that is not fully executable due to a trivial syntax error, wrong import, or misplaced semicolon, yet still be highly valuable because the structural edits, naming, organization, and reasoning align with the user's goal.
>
> ES complements, not replaces, execution. Both are under-approximations, but they under-approximate different behavioral subspaces: tests cover IO-level behaviors; set of references captures edit-level behaviors.
>
> Execution-based correlation on executable benchmarks is a standard way to ask: "When execution is possible, which static metric best predicts success?" ES shows better correlation, demonstrating its superiority to other static measures.
>
> **"Only solves invariance to shared context" and "solves the wrong problem."**
>
> This mischaracterizes our contribution. Invariance to shared context is not incremental; it is foundational. BLEU and similar metrics fail so severely here that they become nearly unusable for revision tasks. ES also solves Property 4 (P4), which requires sensitivity to changes in O. No pairwise metric can satisfy P4. Together, P3 and P4 formalize the distinction between inherited content and actual applied edits. A revision metric must ignore the former and reward the latter.
>
> As for "solving the wrong problem," syntactic similarity is not a wrong problem; it is a necessary complementary signal.
>
> **Tree Edit Distance.**
>
> TED does not satisfy invariance to shared context; normalized TED is dominated by unchanged AST subtrees and, being pairwise, cannot satisfy P3--P4. TED is also expensive (often O(n^3) or O(n^4) [1,2]). Our Related Work already discusses pairwise metrics, and we will explicitly note TED there and add it as a baseline. Our evaluation already includes CodeBLEU [3], which incorporates AST structure. ES outperforms it, underscoring the importance of removing shared context. RUBY [4], based on PDG edit distance with TED fallback, has been shown to perform poorly relative to MT-based metrics [5].
>
> **Ignoring embedding-based metrics in evaluation**
>
> We will add CodeBERTScore. However, it is pairwise and cannot satisfy P3--P4, ignores O, lacks interpretability, and is computationally heavy. ES is lightweight, O-aware, and interpretable.
>
> **Q1: Real-world uses for ES.**
>
> We first started thinking about ExcisionScore in another project that involved evaluating LLMs' ability to de-obfuscate code. A similar problem arises in [6]. TED failed because it rewarded shared context and required syntactic correctness; pairwise semantic metrics lacked interpretability and ignored O. Beyond that initial use case, ES is immediately applicable to:
>
> * **Non-executable or uncompilable revisions.**
> * **Cheap online metrics for human-in-the-loop editing.** ES can track how close LLM suggestions are to what users commit.
> * **Reward models and RLHF.** ES provides stable feedback for partial or invalid outputs.
> * **Cross-language or low-tooling settings.** TED needs parsers; ES only needs tokenization.
> * **Detecting whether two revisions apply the same edit.** ES isolates the actual edit.
> * **Automated program repair.** APR systems generate many uncompilable patches; ES can score them.
> * **Revision-based NLP tasks.** Many tasks modify large documents via small edits. ES applies beyond code.
>
> In summary, ES fills an unmet need for a lightweight, language-agnostic, interpretable, O-aware revision metric that works even when code is invalid, untestable, or non-executable, and when pairwise syntactic or semantic metrics fail to capture meaningful edits.

---

> > ### Author Response · Authors · 2025-11-26
> >
> > **References**
> >
> > [1] http://tree-edit-distance.dbresearch.uni-salzburg.at/
> >
> > [2] An Optimal Decomposition Algorithm for Tree Edit Distance. Erik D. Demaine, Shay Mozes, Benjamin Rossman, and Oren Weimann. https://erikdemaine.org/papers/TreeEdit_ICALP2007/paper.pdf
> >
> > [3] Ren, Shuo, Daya Guo, Shuai Lu, Long Zhou, Shujie Liu, Duyu Tang, Neel Sundaresan, Ming Zhou, Ambrosio Blanco, and Shuai Ma. "CodeBLEU: a method for automatic evaluation of code synthesis." arXiv preprint arXiv:2009.10297 (2020).
> >
> > [4] Ngoc Tran, Hieu Tran, Son Nguyen, Hoan Nguyen, and Tien N. Nguyen. 2019. Does BLEU score work for code migration? In Proceedings of the 27th International Conference on Program Comprehension (ICPC '19). IEEE Press, 165–176. https://doi.org/10.1109/ICPC.2019.00034
> >
> > [5] Mikhail Evtikhiev, Egor Bogomolov, Yaroslav Sokolov, and Timofey Bryksin. 2023. Out of the BLEU: How should we assess quality of the Code Generation models? J. Syst. Softw. 203, C (Sep 2023). https://doi.org/10.1016/j.jss.2023.111741
> >
> > [6] Cao, Qi, et al. "Unnatural error correction: Gpt-4 can almost perfectly handle unnatural scrambled text." arXiv preprint arXiv:2311.18805 (2023).
> >
> > [7] Zheng, Jiasheng, et al. "Beyond correctness: Benchmarking multi-dimensional code generation for large language models." arXiv preprint arXiv:2407.11470 (2024).

---

### Official Review · Reviewer_SvX1 · 2025-10-31

**Soundness:** 3
**Presentation:** 2
**Contribution:** 2
**Rating:** 2
**Confidence:** 4

**Summary:**

The paper introduces a new metric called Excision Score designed to evaluate the quality of edits in documents, such as code or text. he authors show that traditional similarity metrics like BLEU and ROUGE are flawed for documents with large common contexts. Excision Score is designed to capture better correlation with human judgement for such a case. The claim is support with empirical experiments.

**Strengths:**

In experiments on code-editing datasets, the paper demonstrates that Excision Score correlates significantly better with actual test execution results (pass/fail) than existing metrics, especially when a large amount of shared context is present.

**Weaknesses:**

While the work is valuable, the contribution may lack the substantiality and novelty expected for a full paper at ICLR.

**Questions:**

None

---

> ### Author Response · Authors · 2025-11-26
>
> We thank the reviewer for their positive assessment of our experiments and for acknowledging that our work is "valuable". However, we respectfully disagree that our contribution lacks "substantiality and novelty".
>
> **1. On Substantiality:**
>
> The weakness of existing metrics in the presence of shared context is not a minor issue; it is a crippling, systematic bias that invalidates their use for the entire "revision similarity" problem class. This class includes critical tasks like code repair, text simplification, and document updating.
>
> Our contribution is substantial because:
>
> * We provide a principled solution. Excision Score isn't just a tweak; it's a new paradigm. The excision step is a conceptual breakthrough analogous to creating a sterile field in surgery --- it allows for a focused evaluation of the critical region (the edit) by systematically removing the confounding signal (the shared context).
>
> * The empirical gain is significant: aligned with HumanEvalFix tests, ES outperforms SARI by 12% (and >21% over BLEU), and under increased shared-context perturbations its gains grow to 20% over SARI and >30% over standard measures. This represents not an incremental gain but a substantial leap, demonstrating that our principled approach translates to dramatically better performance in practice.
>
> **2. On Novelty:**
>
> The novelty is not merely in using LCS, but in the novel formulation and application:
>
> * Formulation: We are the first to rigorously formulate the revision similarity problem and establish adequacy criteria that any metric for this task must satisfy.
>
> * Application: The specific 3-way LCS excision process, followed by the application of SARI only on the divergent regions, is a novel methodological construct. It provides a general-purpose framework for isolating and evaluating edits, which we believe will influence future metric design beyond this paper.
>
> In summary, we have moved beyond pointing out a weakness in existing metrics to providing a new, principled foundation for evaluating revisions. Given the ubiquity of revision tasks in AI, we are confident that this work represents a substantial and novel contribution worthy of a full paper at ICLR.

---

### Author Response · Authors · 2025-11-26

We thank the reviewers for the time they took to provide feedback. They have helped us see how to better position and strengthen our work.

All of them raised concerns about ExcisionScore's relation to execution. Reflecting on their feedback, we realised that we needed to spend more time on comparing and contrasting the two as well as clarifying our primary use case. Our goal is not to claim superiority where execution is feasible and perfect, but to establish ES as an essential tool for the widespread scenarios where execution is not an option.

The fundamental value of a static metric is greatest precisely when dynamic metrics fail completely: that is, when there is no executable program to evaluate. This is the norm, not the exception, in active software development, where developers constantly work with incomplete, non-compilable code for which tests have not yet been written. Even for compilable code in large systems, full build and test cycles can be prohibitively long, making rapid, lightweight static feedback essential. In these contexts, ES provides a significantly more reliable proxy than previous static metrics like BLEU.

Furthermore, we position ES as a complement to, not a replacement for, dynamic measures. Both are under-approximations. Testing refines its under-approximation by adding more test cases; ES refines its by adding more reference revisions. We contend that because these references are code snippets representing spaces of possible encodings, not single input-output pairs, they under-approximate a different subspace of program behavior. This synergy underscores why demonstrating ES's strong correlation with execution on *executable* benchmarks is a standard and pragmatic validation of its utility as a proxy, even as its primary application remains the vast and critical domain of non-executable code.

In many evaluation settings, metrics based on lexical similarity or semantics and execution act as inexpensive proxies for human preference. We argue, however, that functional correctness --- validated through execution --- does not perfectly align with human judgment, and therefore cannot render static or lexical metrics obsolete. In the increasingly common task of asking an AI assistant to edit code, a functionally correct solution is often insufficient: humans also care about clarity, maintainability, style, idiomatic usage, and alignment with their intent. LLMs frequently violate these expectations. They may introduce excessive or irrelevant comments, follow outdated or non-idiomatic conventions, or produce code that is technically correct yet unreadable --- for example, compressed one-liners, overly clever solutions, or unnecessary abstractions. Conversely, a model may generate code that is not fully executable due to a trivial syntax error, wrong import, or misplaced semicolon, yet still be highly valuable because the structural edits, naming, organization, and reasoning align with the user's goal. In short, neither lexical similarity nor functional correctness alone captures the full spectrum of human judgments in code editing tasks; instead, they provide complementary signals. While execution-based evaluation is mature, there remains a clear unmet need for rigorous static metrics to assess these human-relevant qualities.

Our discussion of property 5 was muddy and too general, verging into undecidability. All of the reviewers picked up on this, to varying degrees. In reponse to the their comments, we have substantially reworked it to clarify its scope to semantically equivalent syntactic variances modulo semantics-preserving transformations. Its purpose to be allow a reference-based measure to better leverage a compact reference set.

We are still working on making the changes needed to address the reviews. We will post a new version by the end of the week.

---

### Author Response · Authors · 2025-12-02
**Updated Manuscript**

We have uploaded the refined manuscript PDF to reflect the reviewers' helpful suggestions.

Summary of the changes (highlighted orange in the PDF):

* We clarified the relationship between ES and execution-based dynamic measures. Introductory paragraph of the Evaluation section now clearly outlines our motivation for basing our evaluation on correlation with tests (questions by dBTr, 9vsX).
* We revised and extended our discussion of Property 5 (Obliviousness to Semantically Equivalent Syntactic Variances). It now includes a rigorous definition and better explains the idea of under-approximating semantics with a reference set (remarks by C6eL).
* Evaluation section now describes the effects of exact matches on the result (suggestion by 9vsX).
* Related work section now explitly mentions TED, CodeBertScore, RUBY (remarks by dBTr).

We are still working on adding TED and CodeBertScore to our evaluation.

---

### Note · Program_Chairs · 2026-01-17
**Submission Desk Rejected by Program Chairs**

The following references in this submission do not refer to real documents and/or have major errors in bibliographic information:

 Christopher Bryant, Mariano Felice, and Ted Briscoe. Building a written corpus of learner english with automatic linguistic annotation. In Proceedings of the 12th Workshop on Innovative Use of NLP for Building Educational Applications, pp. 61-68, 2017.